# Simulation Analysis and Experimental Verification of Freezing Time of Tuna under Freezing Conditions

**Yilin Huo** [1,2,3] , **Dazhang Yang** [1,2,3,*] , **Jing Xie** [1,2,4,*] and **Zhikang Yang** [1,2,3]

1   College of Food Science and Technology, Shanghai Ocean University, Shanghai 201306, China;
    huoyilin10@163.com (Y.H.); m210300783@st.shou.edu.cn (Z.Y.)
2   Shanghai Professional Technology Service Platform on Cold Chain Equipment Performance and Energy
    Saving Evaluation, Shanghai 201306, China
3   National Experimental Teaching Demonstration Center for Food Science and Engineering,
    Shanghai Ocean University, Shanghai 201306, China
4   Quality Supervision, Inspection and Testing Center for Cold Storage and Refrigeration Equipment,
    Ministry of Agriculture, Shanghai 201306, China
*   Correspondence: ydz201@163.com (D.Y.); jxie@shou.edu.cn (J.X.)

**Abstract:** AbstractsIn order to predict the regular temperature change in tuna during the freezing process for cold chain transportation, improve the quality of frozen tuna, and reduce the energy consumption of freezing equipment, a three-dimensional numerical model for freezing tuna of different sizes was established. An unsteady numerical simulation of the air velocity and flow field was combined with an analysis of the freezing process of tuna. This paper also studied the effect of air velocity, temperature, and tuna size on the freezing process. The numerical results show that there was a positive correlation between the cold source environment and the tuna-freezing process. Lower temperatures and higher air increased the velocity at which the tuna moved through the maximum ice crystal formation zone, maintaining a better aquatic product quality. In some cases, however, the smaller tuna models achieved a longer freezing time. Due to the difficulty of obtaining the whole tuna sample, the temperature curve and freezing rate over time obtained during the freezing process were tested using a tuna block of a specific size. The maximum error did not exceed 6.67%, verifying the authenticity and feasibility of the simulation.

**Keywords:** refrigeration systems; CFD; freezing rate; quality; tuna; temperature field

**Key Contribution:** A system has been built to simulate the freezing process of tuna by combining the flow field and the temperature field, and the accuracy has been verified via experiments.





## 1. Introduction

The quality of frozen food, especially aquatic products, is affected by the whole process of food cold chain transportation, which includes the pretreatment, freezing, storage, and transportation of frozen food [1]. Freezing can effectively inhibit the activities of microorganisms in aquatic products, reduce enzyme activity, and effectively prolong the shelf life of frozen foods [2–4]. Taking the freezing process of tuna as an example, due to its perishable characteristics and in order to prolong its shelf life, its minimum temperature needs to be lowered to below −18 °C as soon as possible after fishing, and the fish then needs to be placed into a freezer to further reduce its temperature to below −60 °C [5].

During the freezing process, along with the decrease in the temperature of tuna, the water will change from the liquid phase to the solid phase, forming ice crystals [6]. Ice crystals cause damage to microbial cells and cause the death of harmful microorganisms in tuna, and at the same time, excessive ice crystals also damage the organizational structure of aquatic products [7]. Therefore, it is worth considering how to increase the freezing rate of tuna to quickly pass through the maximum ice crystal formation zone and reduce the

number of large ice crystals formed, which would lead to reduced negative impact of ice crystals on aquatic products [8].

The freezing methods of aquatic products can be mainly divided into three categories according to the freezing speed, namely, fast freezing (5–20 cm/h), medium-speed freezing (1–5 cm/h), and slow freezing (0.1–1 cm/h) [9]. There are differences in the freezing air velocity; the temperature; and the size, shape, and distribution of the ice crystals. As one of the important freezing technologies, rapid freezing can quickly reduce the temperature of the food center to below −18 °C to achieve the purpose of quickly freezing food and reducing cell and muscle damage [10–12].

In order to explore the effect of freezing temperature on the quality of big-eye tuna, Jiang et al. [13] tested factors such as color difference and oxidation stability of tuna frozen at different temperatures and found that the high freezing rate brought about by −40 °C and lower temperatures could effectively inhibit the discoloration and protein changes in aquatic products, and the factors of energy saving were taken into account. However, the study did not explain the difference in freezing rate and quality caused by the freezing of different sizes of tuna. Zhao et al. [14] tested samples of frozen golden pomfret at different temperatures and freezing rates and confirmed that a faster freezing rate was an effective method to reduce the effect on aquatic product quality, but the experiment failed to provide a numerical method to predict the freezing rate and process of fish.

To predict the freezing process of bayberry and the effect of freezing on the quality of it, Zhao et al. [15] conducted a numerical simulation of the freezing process of bayberry and tested the freezing speed and quality of bayberry at different temperatures, but did not consider the cold air velocity during freezing. Wan et al. [16] conducted a numerical simulation of frozen shrimp in refrigerators and predicted the temperature distribution and freezing time, but their study lacked experimental verification and analysis of the effects of freezing conditions on shrimp quality.

Therefore, it is significant to study the freezing rate of tuna in different types of freezing equipment to explore the effects of freezing temperature and air velocity on the quality of aquatic products and to find more energy-efficient freezing methods [17,18]. In this paper, tuna is used as the research object, and different freezing methods are tested. With reference to the equipment temperature and air velocity of the refrigerated equipment, the freezing rate of tuna is analyzed through numerical simulation, and the rationality of the model is verified via experiments. The purpose of this paper is to provide a theoretical reference for studying temperature distribution during the freezing process and for predicting freezing time, to explore reasonable freezing conditions that enable tuna to pass the maximum ice crystal formation zone more quickly, and as well as to explain how to increase the freezing rate to ensure quality while ensuring low energy consumption.

- $\omega$—tuna water content
- $\lambda$—interface thermal conductivity
- $\lambda_l$—interface liquid-phase thermal conductivity
- $\lambda_s$—interface solid-phase thermal conductivity
- $\rho$—tuna density
- CFD—computational fluid dynamics
- $Cp$—tuna specific heat capacity
- $Cp_l$—specific heat capacity before freezing
- $Cp_s$—specific heat capacity after freezing
- $h$—specific enthalpy
- $h_s$—heat transfer coefficient of the tuna surface
- $k$—thermal conductivity of tuna
- $n$—outer normal line direction
- $T$—temperature
- $T_0$—initial temperature
- $T_d$—bottom surface temperature
- $T_{ext}$—air temperature

- $T_l$—phase interface liquid-phase temperature
- $T_p$—phase interface phase transition temperature
- $T_s$—phase interface solid-phase temperature
- $T_{So}$—solidus temperature
- $T_w$—surface temperature
- $T_{fp}$—phase transition temperature
- $t$—freezing time
- $Q$—phase transition enthalpy

## 2. Materials and Methods

### 2.1. Sample Preparation

Yellowfin tuna from the Pacific Ocean was slaughtered and cut into pieces (with each piece weighing about 500 g) after being caught. Samples were stored after vacuum-freezing at −55 °C. Tuna samples were purchased from Zhengyangfan (Xiamen, China). After being thawed to normal temperature (5 °C), the tuna samples were cut into pieces that were 50 × 30 × 10 mm in size and divided into 5 groups.

### 2.2. Water Content

Before the experiment, the tuna samples were dried and chopped to make surimi, which was then weighed in a weighing bottle. The water content of the initial sample was measured using a direct drying method to calculate the thermal property parameters [19]. The temperature of the drying oven was adjusted to 100 °C. The weighing bottle was placed in the drying oven with the bottle cap tilted so that it was to the side of the bottle opening. The sample was heated for 1.5 h and left for half an hour, and then the above operation was continued until the quality difference after two drying sessions was less than 2 mg. The water loss after drying was compared with the total mass to determine the water content of the tuna sample. After the above operation was repeated three times, the average value was taken, and the water content obtained during the simulation calculation was determined to be 73%.

### 2.3. Temperature Curve

A 2640A network multipoint temperature collector (Fluke, Shanghai, China) was connected to a computer before the freezing experiment, and the TT-T-36 calibrated thermocouple temperature probe (Omega Engineering. Inc., Connecticut, America) was fixed at the central point of the tuna block, with the collection interval set at 10 s [20,21]. After the temperature of the refrigerator was adjusted to −18 °C and kept empty for one day before the experiment, the temperature of each point in the freezer was measured with a thermocouple thermometer to ensure that the temperature remained constant at −18 °C.

Based on the above steps, the tuna blocks with fixed thermocouples were taken out of the constant temperature and humidity device and placed in the middle of the refrigerator freezer, making sure that the refrigerator door was rapidly closed, so as to obtain the freezing temperature curves of the center points of tuna pieces under the freezing environment of a temperature of −18 °C and an air velocity of 0 m/s. Because the size of the tuna block is small, the experimental measuring point is not easily fixed; thus, the above steps needed to be repeated to reduce the error. The results show that the temperature curve of the tuna piece at this point is basically the same under the same conditions.

## 3. Models and Assumptions

### 3.1. Physical Models

FLUENT was used to simulate the freezing process of tuna, and the changes in the temperature field and basic parameters of tuna under different air velocities, freezing temperatures, and fish meat sizes were calculated. In order to obtain the model with the highest freezing efficiency, this paper added other sizes on the basis of simulating the freezing process of the whole tuna fish.

The structural parameters are: (1) 0.6 m for a whole fish; (2) 0.3 m for fish segment A (taking the widest part of the tuna); (3) 0.15 m for fish segment B (taking the widest part of the tuna); (4) 100 × 50 × 30 mm for piece A; and (5) 50 × 30 × 10 mm for piece B. The tuna is in direct contact with the surface of the freezing device, as shown in Figure 1. During freezing, cold air flows in from the left side of the model to perform convective heat transfer. After heat exchange, the cold air flows out in other directions under the action of wind pressure.

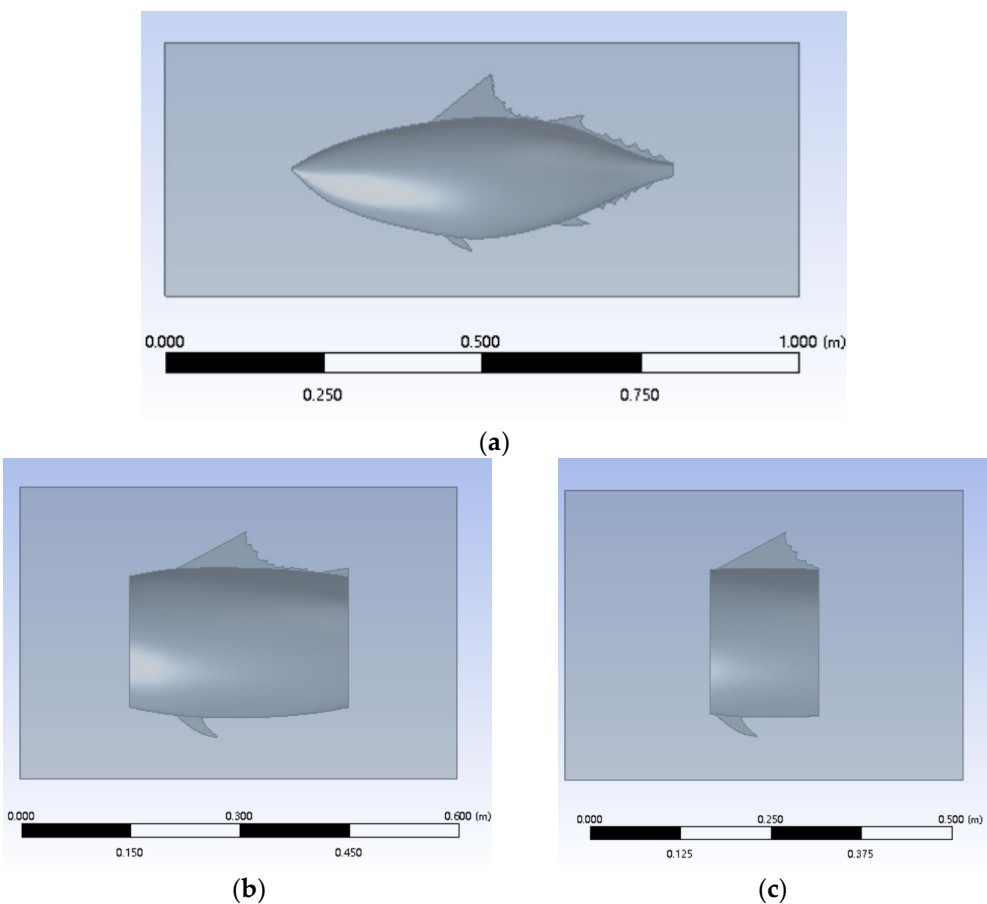

**Figure 1.** Physical models of tuna of different sizes: (**a**) whole fish; (**b**) fish segment A; (**c**) fish segment B.

### 3.2. Mathematical Models

Since the freezing process of tuna is simulated in this paper, and the process involves a phase transition, the thermophysical property parameters required for the simulation mainly include the specific heat capacity, phase transition enthalpy, phase transition temperature, thermal conductivity, and density of the tuna. For the freezing process of most food products, the phase transition does not occur at a definite temperature, but within a relatively small temperature range. The thermophysical parameters such as the thermal conductivity of tuna will change before and after the phase transition. Therefore, the difference in physical parameters before and after freezing the tuna cannot be ignored. In this simulation, the physical parameters such as specific heat capacity during the freezing process of tuna are calculated using formulas that refer to the water content and phase transition, as provided by Gong et al. [22]. Data, such as enthalpy and phase transition temperature data, were used to determine the physical properties of the tuna, as shown in Table 1.

**Table 1.** Thermophysical parameters of tuna.

| Parameters | Before Freezing | | After Freezing | |
|---|---|---|---|---|
| | Value | Calculated with | Value | Calculated with |
| $Cp$ (kJ/(kg·K)) | 3.282 | $Cp_l = 0.837 + 3.349\omega$ | 1.754 | $Cp_s = 0.837 + 1.256\omega$ |
| $\lambda$ (W/(m·K)) | 0.508 | $\lambda_l = 0.26 + 0.34\omega$ | 1.519 | $\lambda_s = 2\omega + 0.22(1-\omega)$ |
| $\omega$ | | 0.73 | | |
| $\rho$ (kg/m$^3$) | | 1050 | | |
| $T_p$ (K) | | 270.1 | | |
| $T_{So}$ (K) | | 264.3 | | |
| $Q$ (J/kg) | | 176,500 | | |

*3.3. Assumptions*

On the basis of ensuring the accuracy of the numerical simulation, our model adopts several assumptions: (1) It is assumed that the initial temperature of the model is uniform, the interior is isotropic, and the boundary conditions are constant. (2) It is assumed that the radiation between different products is not included, and only the changes that occur in a single aquatic product during freezing are predicted. Since the radiation temperature between different products in the model was roughly the same, the radiative heat exchange was negligible compared to that resulting from convection. (3) It is assumed that the heat transfer through the wall is negligible, since the wall is considered to be highly insulating due to having a thick insulating layer. (4) It is assumed that moisture evaporation and condensation are negligible [23]. (5) The gas flow and heat transfer predictions are based on the laws of mass momentum and energy conservation, assuming that the gas inside the model follows the ideal gas law.

**4. Boundary Conditions and Calculation Methods**

Gambit software was used to couple the contact surface between the fluid and tuna in the model. Then, the model was meshed, and the FLUENT software was used to perform numerical simulation calculations on the meshed model. The boundary conditions of the model simulation process were set according to the actual cold chain transportation conditions.

*4.1. Inlet and Outlet*

In the simulation, the inlet air velocity of the refrigeration equipment was 0 m/s, 1 m/s, 6 m/s, and 20 m/s, and the temperature was −18 °C, −35 °C, and −60 °C. The turbulence is defined by the strength and characteristic of the inlet; here, the turbulence intensity was 5%. To simplify the calculations, the outlet was set as the outflow boundary.

*4.2. Boundary Conditions*

Since the assumption ignores the radiative heat exchange between aquatic products, the freezing process of tuna is mainly realized by the convective heat exchange between the cold air and the surface of the tuna [24,25].

4.2.1. Internal Heat Conduction Equation

Based on the above premises, the freezing process of tuna is regarded as a variable physical property and unsteady heat conduction problem without an internal heat source [26,27]:

$$\rho(T)Cp(T)\frac{\partial T}{\partial t} = \nabla(k(T)\nabla T) + h_s(T)(T_w - T_{ext}) \tag{1}$$

4.2.2. Bottom Equation

The inner wall of the freezing device is insulated to avoid heat transfer. The bottom of the tuna is in direct contact with the wall of the freezing equipment. The temperature

of the wall is assumed to be equal to that of the cold air. The temperature of the bottom portion of the tuna was approximately the same as that of the wall:

$$T_d = T_{ext} \tag{2}$$

The initial condition is $T_d = T_0$, where $T_0 = 278.15$ K is the initial temperature of the tuna.

Surface Convection Equation

When the air speed at the air inlet is 0 m/s, natural convection occurs between the sides of the tuna and the cold air in the freezing device. When there is air velocity at the air inlet, forced convection occurs between the tuna and the air in the freezing device, which is regarded as the third boundary condition [28]:

$$-k(T)\frac{\partial T}{\partial n} = h_s(T)(T_w - T_{ext}) \tag{3}$$

Phase-Surface Equation

For the mobile-phase interface, the model satisfies the law of mass conservation and energy conservation:

$$T_s(x, y, z, t) = T_l(x, y, z, t) = T_p \tag{4}$$

$$\lambda_s \left( \frac{\partial T_s}{\partial x} + \frac{\partial T_s}{\partial y} + \frac{\partial T_s}{\partial z} \right) - \lambda_l \left( \frac{\partial T_l}{\partial x} + \frac{\partial T_l}{\partial y} + \frac{\partial T_l}{\partial z} \right) = h\rho \left( \frac{dx}{dt} + \frac{dy}{dt} + \frac{dz}{dt} \right) \tag{5}$$

*4.3. Calculation Method*

The simulations were modeled using SolidWorks 2021 and meshed in ANSYS 2021R1. The element order of the model was set to linear, and the inflation option was adjusted for smooth transitions. In order to prove that the simulation results are independent of the number of grids, the calculation was carried out with grids of different sizes. After testing models with elements of different orders of magnitude, it was found that different mesh divisions caused small differences in the simulation. Therefore, it can be assumed that when each model selects the current mesh division method, the calculation results have nothing to do with the mesh number. The mesh quantity details and independence analysis are shown in Table 2.

**Table 2.** Mesh quantity details and independence analysis of different models.

| Parameters | Whole Fish | Segment A | Segment B | Piece A | Piece B |
|---|---|---|---|---|---|
| Mesh quality | | | | | |
| Nodes of fish | 3301 | 3990 | 3766 | 3828 | 2496 |
| Elements of fish | 14,820 | 18,435 | 17,042 | 10,924 | 1875 |
| Nodes of air | 12,540 | 15,180 | 13,949 | 14,473 | 15,676 |
| Elements of air | 68,863 | 79,347 | 74,969 | 72,677 | 81,951 |
| Independent analyses | | | | | |
| Order of magnitudes | Freezing time at −35 °C and −6 m/s air velocity (s) | | | | |
| $10^4$ | 25,285 | 25,487 | 18,793 | 2954 | 574 |
| $10^5$ | 26,685 | 27,254 | 20,905 | 3408 | 702 |
| $10^6$ | 27,454 | 28,057 | 21,526 | 3479 | 731 |

The left air outlet was set to be the velocity inlet, and the air supply temperature was set according to three different experimental groups: (1) $-18\,^{\circ}$C (the temperature for general food cold chain transportation); (2) $-35\,^{\circ}$C (the temperature of the fast-freezing equipment); and (3) $-60\,^{\circ}$C (the ultra-low temperature freezing temperature required for transporting tuna) [29–31].

The air supply speed was set according to three different types of experimental groups: (1) 0 m/s (the air velocity of general direct-cooling refrigeration storage); (2) 1 m/s (the air velocity of air-cooling refrigeration storage and household air-cooling refrigerators); (3) 6 m/s (the air velocity of spiral-type quick-freezing equipment); and (4) 20 m/s (air velocity of impact-type quick-freezing equipment) [32,33].

The temperature at the bottom of the model and the air remains constant, and thus, the air outlets on the other four sides were set as the outflow boundary. The surface of the tuna is the coupling surface, and the software automatically performs the coupling calculation according to the set physical parameters. For the initial setting, the temperature of the tuna was 278.15 K, and the ambient temperature in the freezing area was consistent with the cold air temperature.

The model was set according to the above parameters, the K-ε model and the SIMPLE solution method were selected, and the pressure-based transient solver was used to solve the model. The initial time step was set to 1 s, and the time step was gradually increased to 5 s after the residual curve converged. The convergence accuracy of the energy equation was set to $10^{-6}$, and the convergence accuracy of the remaining equations was $10^{-3}$. The simulation ended when the maximum temperature of the tuna was detected to be lower than 255.15 K.

## 5. Results and Discussion

### 5.1. Accuracy of Numerical Simulations

The freezing curve of the tuna piece at the central point under conditions of $-18\,^{\circ}$C and a 0 m/s air velocity was measured and compared with the simulated value. From the freezing curve of the tuna, it can be seen that the freezing process is divided into three stages [34,35]. In the first stage, the temperature decreases rapidly to the freezing point, and the freezing rate decreases gradually, which might be due to the decrease in the temperature difference between the tuna surface and the environment after heat exchange occurs. In the second stage, the temperature at the center of the tuna reaches the range of the phase transition, and latent heat is released during the phase transition, which greatly reduces the freezing rate and makes the temperature curve smooth [36,37]. In the final stage, the water inside the tuna freezes, and the cooling rate is lower than that at the initial stage because the specific heat of the ice is lower than that of water [38].

From the comparison between the simulated values and the experimental values in Figure 2, it can be seen that the simulated values at the beginning and end of the freezing process have a higher degree of harmony with the experimental values, and there is a certain difference in their temperature curves as the freezing enters the phase transition period. This may be due to the difference between the phase transition temperature in the simulation and the actual phase transition temperature in the experimental process [39]. After calculation, the correlation coefficient between the simulated freezing time curve and the experimental curve is 0.995, and the error between the simulated and actual curve is 6.67%, which shows that this model can accurately predict the freezing time of tuna.

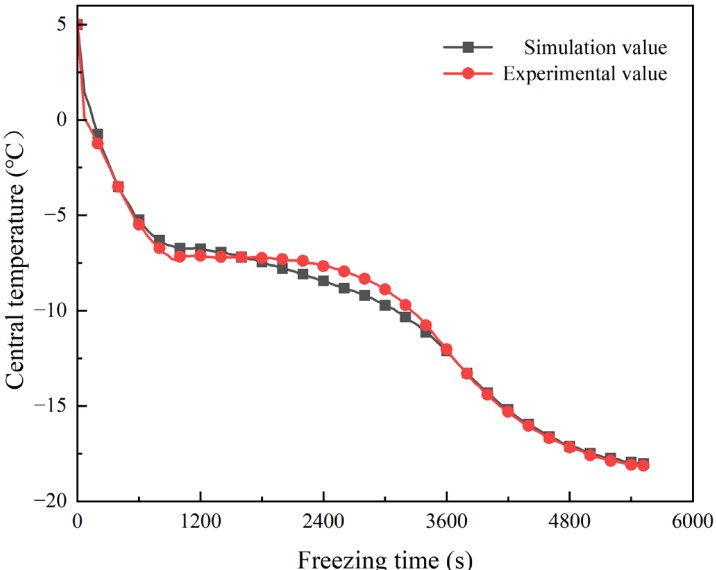

**Figure 2.** Comparison of experimental and simulated values of central temperature of tuna.

### 5.2. Analysis of Velocity and Temperature Field

The air velocity of the refrigeration equipment used in cold chain transportation is not 0 m/s. Therefore, let us take the example of freezing the tuna using cold air at a temperature of −60 °C and an air velocity of 1 m/s. Under this condition, the temperature field inside the tuna and the velocity field of the environment are shown in Figure 3. In the temperature field, the tuna meat near the bottom is cooled faster due to the direct contact between the bottom of the tuna and the cooling device. The temperature difference between different areas due to the above reasons can be improved with a faster air velocity and lower temperature.

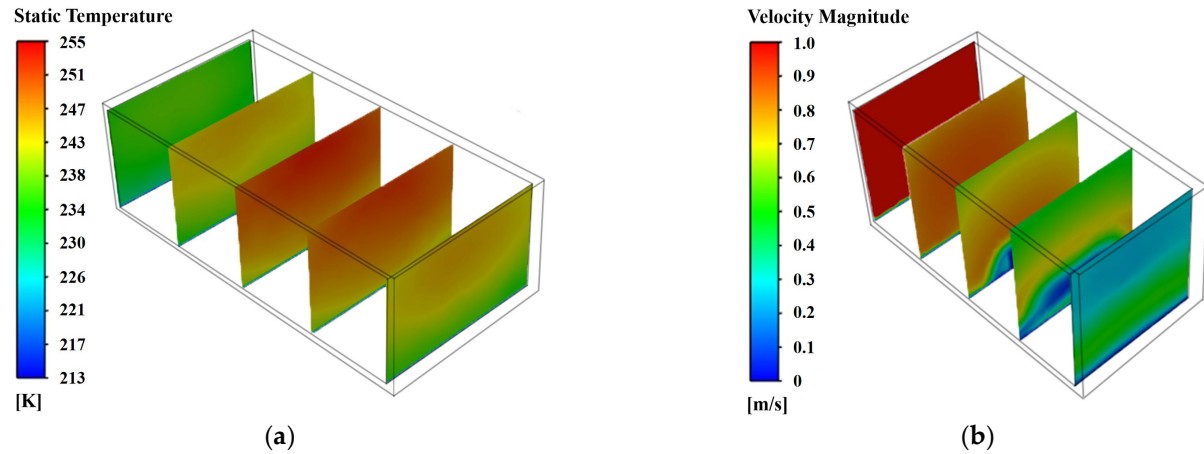

**Figure 3.** Nephogram of the temperature field inside tuna and the velocity field of the environment under specific conditions: (**a**) nephogram of temperature distribution; (**b**) nephogram of velocity distribution.

When the air hits the 100 × 50 × 30 mm tuna block, the flow velocity in the area blocked by the tuna decreases rapidly, while the part close to the tuna wall gains a higher air velocity due to the increased flow. The cold air flows from left to right along the surface of the tuna, and after the heat exchange, the air flows back to the right, so the air velocity around the right part of the tuna is slightly higher than that around the whole tuna block. At the same time, because the gas density increases in a low-temperature environment, and the bottom of the tuna block clings to the wall of the refrigeration equipment that maintains

a low temperature, the air tends to flow downward, and the air velocity on the lower side is higher.

### 5.3. Analysis of Temperature Curve and Nusselt Number

The freezing process of the whole fish was analyzed under an air velocity of 0 m/s, 1 m/s, 6 m/s, and 15 m/s and a freezing temperature of −18 °C, −35 °C, and −60 °C. Figure 4 compares the temperature curve of the highest point of the tuna temperature and the Nusselt number of the tuna surface under different conditions. According to the comparison chart, it can be found that air velocity and temperature have a significant effect on the freezing time.

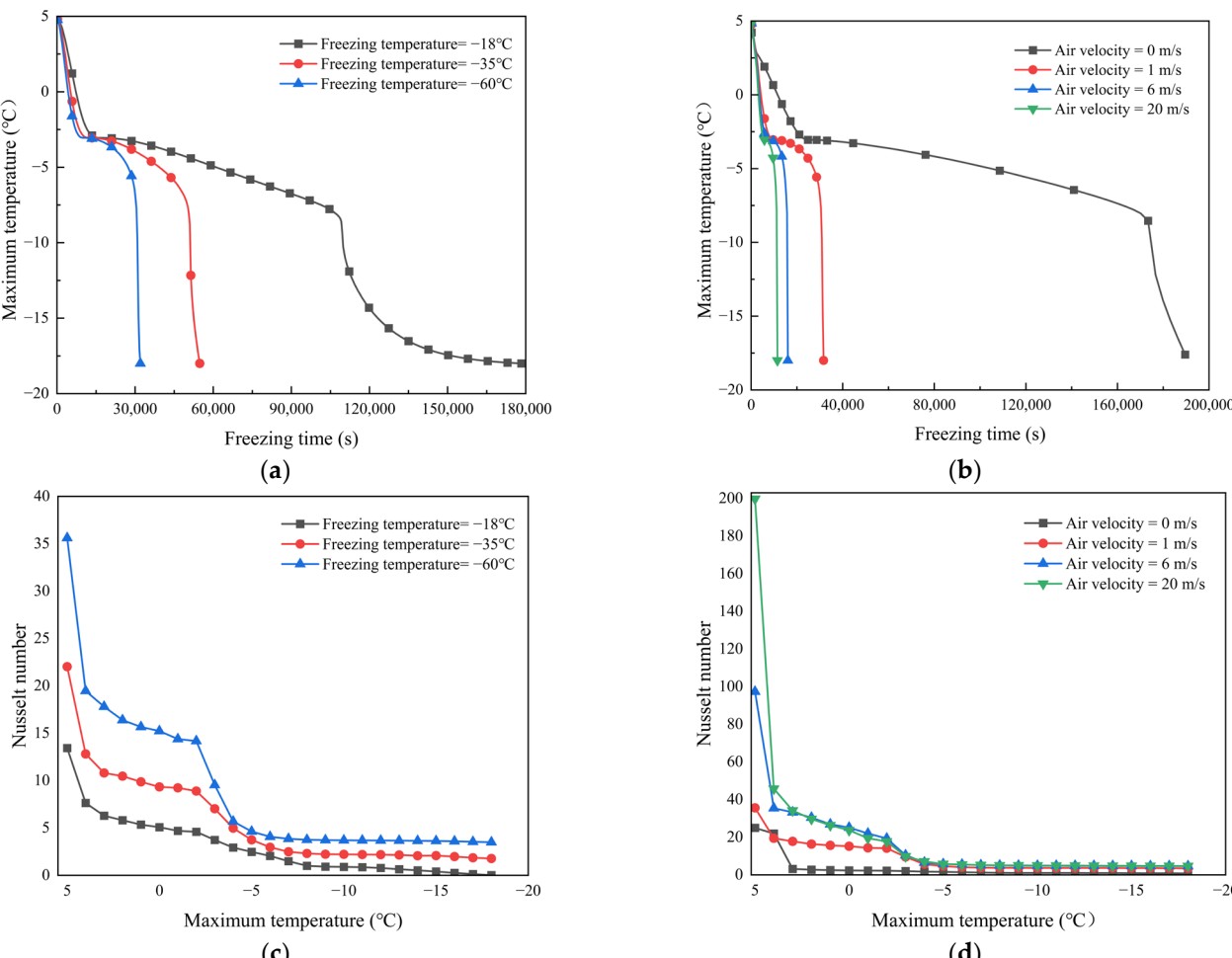

**Figure 4.** Maximum temperature and Nusselt number under different conditions: (**a**) maximum temperature curve of tuna with 1 m/s air velocity; (**b**) maximum temperature curve of tuna with −60 °C freezing temperature; (**c**) Nusselt number with 1 m/s air velocity; (**d**) Nusselt number with −60 °C freezing temperature.

When the air velocity during the freezing process is fixed, the lower temperature can help the tuna complete the phase transition more quickly and lower it to the temperature needed for storage. When the freezing temperature decreases, the convective heat transfer coefficient will develop in the opposite direction, and the Nusselt number will increase [40].

### 5.4. Effect of Air Velocity on Freezing of Tuna of Different Sizes

Using varying air velocities, simulations were carried out on tuna samples of different sizes under different temperature environments. The freezing time required to reduce the

maximum temperature of the tuna model to −18 °C under different conditions is shown in Figure 5.

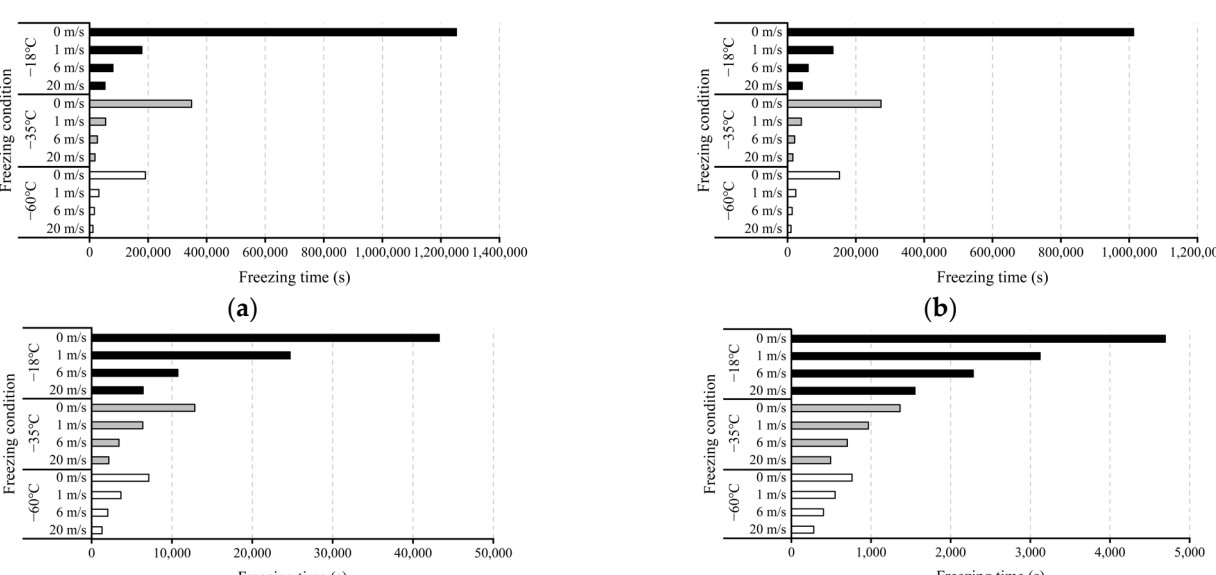

**Figure 5.** Effect of air velocity on freezing time of tuna of different sizes at different temperatures: (**a**) 0.6 m, fish; (**b**) 0.15 m, segment B; (**c**) 100 × 50 × 30 mm, piece A; (**d**) 50 × 30 × 10 mm, piece B.

According to the histogram of the freezing time of tuna under different conditions, under the premise of other conditions being equal, the freezing time of tuna without forced convection in the direct-cooling refrigeration equipment is much higher than that when the tuna is frozen in the impact freezing equipment at 20 m/s. The freezing time of natural convection can even reach 20 times that of using an impact freezing device, and this difference is compensated for by the lower-temperature environment. However, the time gap between using 6 m/s air-cooling cold storage and using 20 m/s impact freezing equipment is not extremely different. When the requirements are not strict, after balancing the freezing speed and the energy saving effect, the 6 m/s air velocity can be used to freeze tuna.

### 5.5. Effect of Temperature on Freezing of Tuna of Different Sizes

Under varying temperatures, simulations were carried out on tuna samples of different sizes under different air velocities. The freezing time required to reduce the maximum temperature of the tuna model to −18 °C under different conditions is shown in Figure 6.

According to the histogram of the freezing time of tuna under different conditions, under the premise of other conditions being the same, when the freezing temperature of tuna is −18 °C, the freezing time when using ordinary cold chain transportation refrigeration equipment is higher than that when using fast-freezing equipment at −35 °C. The freezing time when freezing at −18°C can be more than three times that of using −35 °C fast-freezing equipment. The time difference between using −35 °C fast-freezing equipment and −60 °C ultralow-temperature freezing equipment is not very different. When the requirements are not strict, after balancing the freezing speed and the energy saving effect, the −35 °C fast-freezing equipment can be used to freeze tuna.

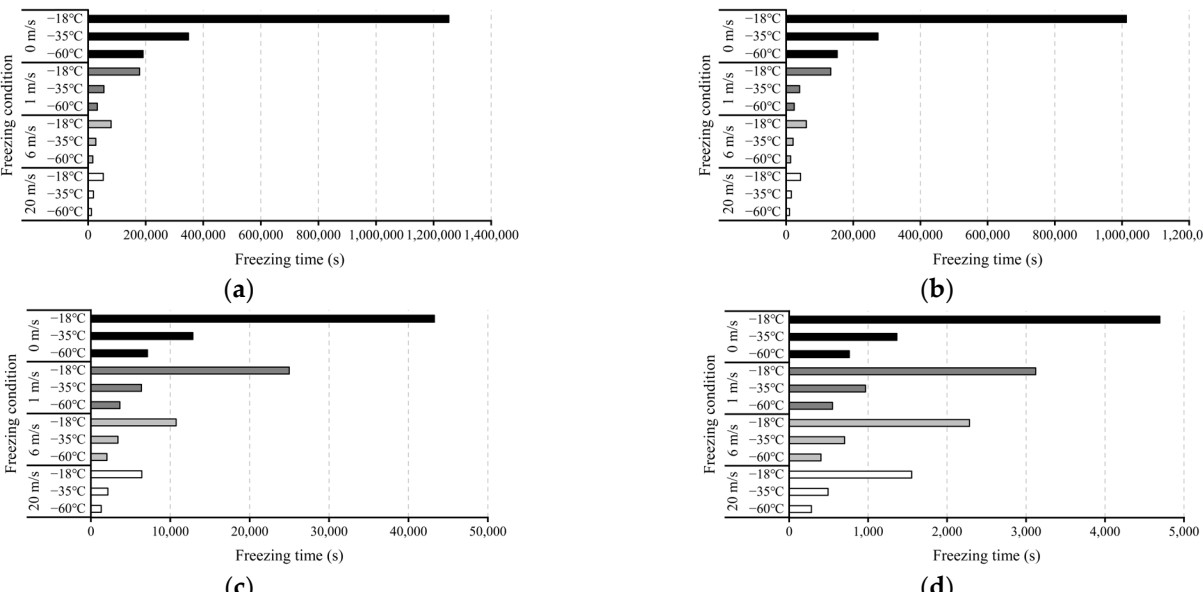

**Figure 6.** The effect of temperature on the freezing time of tuna of different sizes under different air velocities: (**a**) 0.6 m, whole fish; (**b**) 0.15 m, fish segment B; (**c**) 100 × 50 × 30 mm, piece A; (**d**) 50 × 30 × 10 mm, piece B.

*5.6. Effect of Fish Size on Freezing of Tuna*

In the case of changing the size of the tuna, simulations were carried out under different air velocity and temperature environments. The freezing time required to reduce the maximum temperature of the tuna model to −18 °C when the tuna is cut into different sizes is shown in Figure 7, and the freezing rate is shown in Table 3. According to the histogram of freezing time for tuna of different sizes, it can be seen that the freezing time of tuna is basically positively correlated with the size of the fish.

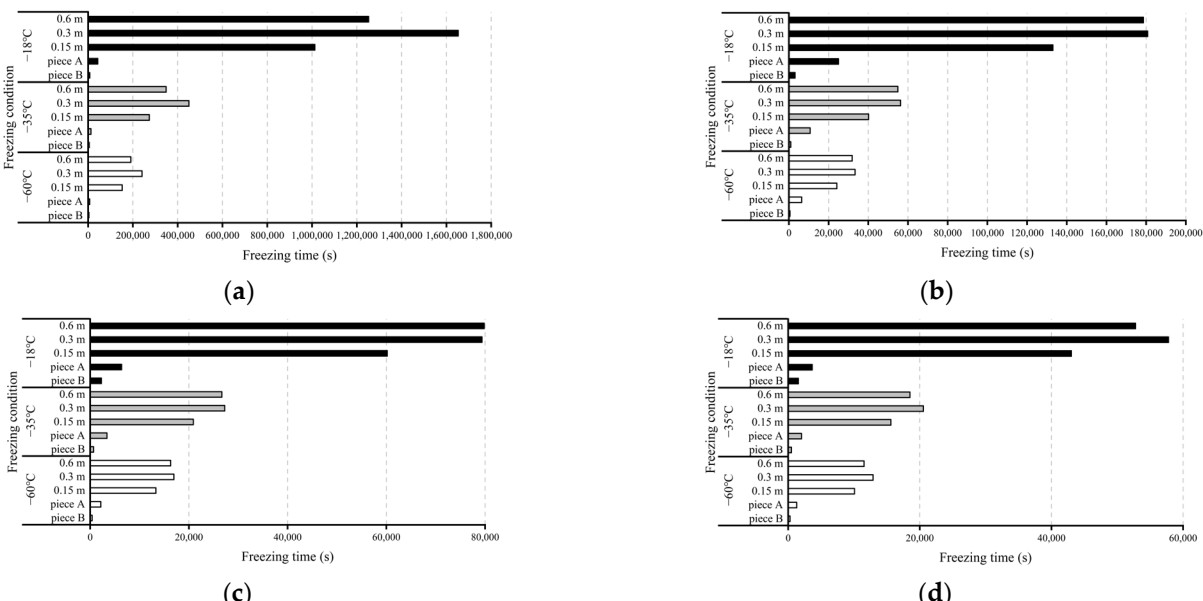

**Figure 7.** The effect of tuna size on freezing time with different air velocities: (**a**) 0 m/s; (**b**) 1 m/s; (**c**) 6 m/s; (**d**) 20 m/s.

**Table 3.** The freezing rate of different tuna sizes and freezing environments.

| Freezing Condition | | Freezing Rate | | | | |
|---|---|---|---|---|---|---|
| Temperature | Air Velocity | Whole Fish | Segment A | Segment B | Piece A | Piece B |
| −18 °C | 0 m/s | 0.020 | 0.015 | 0.024 | 0.570 | 5.255 |
| | 1 m/s | 0.138 | 0.136 | 0.185 | 0.998 | 7.901 |
| | 6 m/s | 0.309 | 0.311 | 0.410 | 2.298 | 10.805 |
| | 20 m/s | 0.467 | 0.427 | 0.573 | 3.845 | 15.894 |
| −35 °C | 0 m/s | 0.071 | 0.055 | 0.090 | 1.921 | 18.084 |
| | 1 m/s | 0.449 | 0.439 | 0.614 | 3.874 | 25.509 |
| | 6 m/s | 0.924 | 0.905 | 1.180 | 7.238 | 35.138 |
| | 20 m/s | 1.333 | 1.201 | 1.581 | 11.484 | 50.137 |
| −60 °C | 0 m/s | 0.129 | 0.102 | 0.162 | 3.464 | 32.372 |
| | 1 m/s | 0.774 | 0.740 | 1.020 | 6.758 | 44.931 |
| | 6 m/s | 1.513 | 1.455 | 1.854 | 12.211 | 61.361 |
| | 20 m/s | 2.138 | 1.912 | 2.447 | 19.048 | 87.784 |

But the exception to this is that when the size of the fish block is 0.3 m, the freezing rate is sometimes slightly higher than that when freezing the whole tuna fish of 0.6 m, which may be the combined effect of the heat exchange area and tuna volume. Because the whole tuna fish can be approximated to be a spindle type, after the size of the tuna is changed, the cooling capacity obtained through the heat exchange surface is reduced even though the heat exchange area of the tuna is reduced, and the cooling capacity is reduced to the required temperature despite not being significantly lowered.

## 6. Conclusions

By taking tuna that is frozen in different freezing equipment as the object, this paper establishes different size models of tuna and discusses the effect of different freezing equipment and conditions on the freezing rate of tuna during cold chain transportation. Through the study of the effects of cold air velocity, freezing temperature, and tuna size on the freezing time using data simulation and experimental verification, the following conclusions have been drawn:

When forced convection refrigeration equipment with an air velocity of 20 m/s is used, the freezing speed is the fastest, and the time required to reduce the maximum temperature of tuna to −18 °C is only one-twentieth of that in an environment without air velocity. Even if an air velocity of only 1 m/s is added in an environment with no air velocity, the freezing efficiency and rate can be significantly improved to reduce the effect of ice crystal formation on the quality of tuna.

The shortest freezing speed can be obtained using ultralow-temperature refrigeration equipment with a temperature of −60 °C, and the time required to reduce the maximum temperature of tuna to −18 °C is only less than a quarter of that of ordinary cold chain transportation equipment at −18 °C. But after evaluating the energy loss and freezing time, the −35 °C flash-freezing equipment can also be used as a good option to reduce the temperature of tuna to −18 °C.

The fastest freezing speed and the shortest freezing time were obtained using smaller tuna pieces. However, the smaller 0.3 m tuna segment has a longer freezing time in some cases than the whole 0.6 m tuna, which is the result of the combined effect of the surface heat transfer area of the tuna and volume of the tuna.

**Author Contributions:** Supervision, D.Y. and J.X.; writing—review and editing, J.X. and Z.Y.; software, Y.H.; investigation, Y.H.; writing—original draft preparation, Y.H.; methodology, D.Y. All authors have read and agreed to the published version of the manuscript.

**Funding:** This research was funded by [Shanghai Professional Technology Service Platform on Cold Chain Equipment Performance and Energy Saving Testing Evaluation] grant number [20DZ2292200]

and [Science and Technology Innovation Action Plan of Shanghai Science and Technology Commission] grant number [19DZ1207503].

**Institutional Review Board Statement:** Not applicable.

**Data Availability Statement:** The data presented in this study are available on request from the corresponding author.

**Conflicts of Interest:** The authors declare no conflict of interest.

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
