# Peer review of "Simulation Analysis and Experimental Verification of Freezing Time of Tuna under Freezing Conditions"

_fishes, doi:10.3390/fishes8090470_

Round 1

Reviewer 1 Report

The manuscript includes a very practical study, focused on the optimisation of freezing conditions of tuna species.

My genera opinion is that the manuscript provides a deep mathematical study, but biochemical aspects have been very scarcely considered.

Some concrete aspects would be as follows:

Abstract

The information provided about what has really been carried out is too scarce. I think the journal allows up to 200 words.

Keywords

Some keywords concerning quality, chemical changes, etc., ought to be included. The study ought to also include the minimisation of protein and lipid changes, for example.

Introduction

Lines 64-66: Again, no indication of quality changes of the product is mentioned.

A possible effect of the composition of the starting tuna is not considered. Great variations of lipid content in pelagic species such as those included in the term “tuna fish” should have an improtant effect on the current study.

Tuna pieces are mentioned, but concrete fish species concerned is/are not mentioned in the text.

Moderate performances could be done.

Reviewer 2 Report

The authors tried to predict the temperature changes of tuna during the freezing process in order to improve the frozen quality of tuna as well reducing the energy consumption during freezing. For this study they used a three-dimentional numerical model of freezing tuna with different sizes. They also studied the effect of air velocity, temperature and size of tuna during freezing process of tuna and then they experimentally verified the reality and feasibility of the simulation.

The manuscript contains some valuable results. However, some of the conclusion drawn from this study has already been reported by the earlier research. For example, high air velocity can lead to faster freezing speed and smaller the size of fish results in faster freezing etc. Therefore, these points are the drawbacks of the manuscript in terms of scientific novelty.

Other issues in the study are as follows;

1. The abstract does not include the results and conclusion. They should be added into the abstract as well.

2. In the conclusion the authors claim that 'The paper takes tuna in different freezing equipment as object, establishes different sizes model of tuna, and discusses the effect of different freezing equipment and conditions on the freezing rate of tuna during cold chain transportation.' Unfortunately, in the method section, the authors did not give different freezing equiptment.

3. The language of the manuscript requires editing.

The language of the manuscript requires editing.

Round 2

Reviewer 1 Report

The authors have included some performances according to previous comments. Thus, several references related to quality in frozen fish have been included. However, comments on such references and aspects are very scarce or even negligible. I think this can be performed in the Introduction and Conclusions sections. I think also the keywords section could be enlarged by including some terms encountered in the study.

Minor performances could be done-
